# Industrial-Scale Production of Mycotoxin Binder from the Red Yeast *Sporidiobolus pararoseus* KM281507

**DOI:** 10.3390/jof8040353

**Published:** 2022-03-30

**Authors:** Wanaporn Tapingkae, Orranee Srinual, Chompunut Lumsangkul, Hien Van Doan, Hsin-I Chiang, Atchara Manowattana, Pinpanit Boonchuay, Thanongsak Chaiyaso

**Affiliations:** 1Department of Animal and Aquatic Sciences, Faculty of Agriculture, Chiang Mai University, Chiang Mai 50200, Thailand; wanaporn.t@cmu.ac.th (W.T.); orranee_sr@cmu.ac.th (O.S.); chompunut.lum@cmu.ac.th (C.L.); hien.d@cmu.ac.th (H.V.D.); 2Innovative Agriculture Research Center, Faculty of Agriculture, Chiang Mai University, Chiang Mai 50200, Thailand; 3Department of Animal Science, National Chung Hsing University, Taichung 40227, Taiwan; samchiang@nchu.edu.tw; 4Faculty of Arts and Sciences, Western University, Kanchanaburi 71170, Thailand; atchara.ma@western.ac.th; 5Division of Biotechnology, Faculty of Agro-Industry, Chiang Mai University, Chiang Mai 50100, Thailand; pinpanit_boonchuay@cmu.ac.th

**Keywords:** 300 L bioreactor, adsorbents, β-glucan, feed additive, gastrointestinal model

## Abstract

Red yeast *Sporidiobolus pararoseus* KM281507 has been recognized as a potential feed additive. Beyond their nutritional value (carotenoids and lipids), red yeast cells (RYCs) containing high levels of β-glucan can bind mycotoxins. This study investigated the industrial feasibility of the large-scale production of RYCs, along with their ability to act as a mycotoxin binder. Under a semi-controlled pH condition in a 300 L bioreactor, 28.70-g/L biomass, 8.67-g/L lipids, and 96.10-mg/L total carotenoids were obtained, and the RYCs were found to contain 5.73% (*w*/*w*) β-glucan. The encapsulated RYC was in vitro tested for its mycotoxin adsorption capacity, including for aflatoxin B1 (AFB1), zearalenone (ZEA), ochratoxin A (OTA), T-2 toxin (T-2) and deoxynivalenol (DON). The RYCs had the highest binding capacity for OTA and T-2 at concentrations of 0.31–1.25 and 0.31–2.5 µg/mL, respectively. The mycotoxin adsorption capacity was further tested using a gastrointestinal poultry model. The adsorption capacities of the RYCs and a commercial mycotoxin binder (CMB) were comparable. The RYCs not only are rich in lipids and carotenoids but also play an important role in mycotoxin binding. Since the industrial-scale production and downstream processing of RYCs were successfully demonstrated, RYCs could be applied as possible feed additives.

## 1. Introduction

Economic loss due to mycotoxin contamination is a global issue [1,2,3]. In the US agricultural economy, a quarter of all crops are damaged by mycotoxin-related issues [4]. The US and Canada have reported significant annual economic loss because of mycotoxins, with losses ranging from USD 1.4 to 5.0 billion per year. Mycotoxin contamination of animal feed is one of the most serious global concerns. The mycotoxins most commonly found in foods and feeds have been determined to be aflatoxins (AF: B1, B2, G1 and G2); zearalenone (ZEA); deoxynivalenol (DON); T-2 toxin (T-2); fumonisins (FUM: FB1, FB2 and FB3); patulin and ochratoxin A (OTA) [5,6,7]. The European Union (EU) has the strongest mycotoxin restrictions in the world. According to the European Food Safety Authority (EFSA), a maximum permitted level of 200 µg/kg for AF and guidance or recommended values of 5000, 250, 250, 100 and 20,000 µg/kg were regulated for DON, ZEA, T-2 and OTA, respectively, for poultry feed. It is difficult to distinguish mycotoxins, because they do not have distinct odors and do not change the organoleptic properties of foods and feeds [8]. Mycotoxin consumption can cause biological effects in animals, including liver and kidney toxicities; neurological, estrogenic and teratogenic effects and immunosuppression [6,9,10]. In general, the visible signs of toxicity are reduced egg and milk production, carcass quality, weight gain, feed conversion and feed intake. At the same time, the likelihood of bloody diarrhea, severe dermatitis, hemorrhage and mortality are increased [4,11] due to secondary contamination [5].

To minimize the harmful effects of mycotoxin contamination, several approaches have been investigated in order to degrade, destroy, inactivate and remove mycotoxins from contaminated feeds using physical, chemical or biological methods [12,13]. Among these, the use of adsorbents to bind mycotoxins is one of the most effective and common strategies against mycotoxin-induced toxicosis [14]. These materials can adsorb or bind mycotoxins, forming a complex substance that limits toxicity [15]. Mycotoxin binders can be categorized into different groups, such as inorganic (e.g., clays, bentonites and aluminosilicates) and organic adsorbents (e.g., yeast (*Saccharomyces cerevisiae*) cells (YCs) and glucomannans) [16,17]. Among these, organic adsorbents have gained more attention than inorganic ones because of their high adsorption efficiency against a broad spectrum of mycotoxins [18]. Recently, YCs have been employed commercially as a mycotoxin binder in the broiler industry [19,20]. Moreover, YCs act as both adsorbents and beneficial feed additives [21].

Red yeast (*Sporidiobolus pararoseus* KM281507) is an oleaginous and carotenogenic yeast. It produces and accumulates carotenoids, a class of isoprenoid compounds, and lipids in the cell [22,23,24]. Moreover, similar to other yeasts, the active components of red yeast cells (RYCs), such as polysaccharides and carotenoids, possess antioxidant, anti-inflammatory, antiapoptotic, antigenotoxicity and anticancer activities [25]. However, several studies have focused on the mycotoxin-binding efficiency of *S. cerevisiae* and its cell wall components [26,27,28]. Carotenoid-producing yeasts, including *Rhodotorula benthica*, *Xanthophyllomyces dendrorhous* (*Phaffia rhodozyma*) and *Sporid. pararoseus*, have been employed as feed additives. Most studies have focused on their bioavailability and the enhancement of pigment formation in feedstocks [24,29,30,31,32]. As the production of red yeast as a feed additive using conventional processes may be insufficient for animal farming, there is a need to investigate the large-scale production of RYCs.

To confirm the industrial feasibility, RYCs were cultivated in a 300 L bioreactor and analyzed for growth kinetics parameters compared with 5 L and 30 L bioreactors. The potential of the RYCs to adsorb five different mycotoxins, namely AFB1, ZEA, OTA, T-2 and DON, was evaluated using enzyme-linked immunosorbent assays (ELISA). In addition, the mycotoxin-binding capacity of the RYCs and a commercial mycotoxin binder (CMB) were evaluated through an in vitro preliminary test and a gastrointestinal poultry model.

## 2. Materials and Methods

### 2.1. Chemicals and Materials

The β-glucan enzymatic assay kit (β-glucan assay kit, yeast and mushroom) was purchased from Megazyme Ltd. (Wicklow, Ireland). The mycotoxin test kit (RIDASCREEN^®^FAST mycotoxin test kit; RIDASCREEN^®^FAST Aflatoxin B1, Ochratoxin A, T-2 Toxin, Zearalenone and DON) and standard mycotoxins (aflatoxin B1, AFB1; T-2 toxin, T-2; ochratoxin A, OTA; zearalenone, ZEA and deoxynivalenol, DON) were purchased from R-Biopharm AG (Darmstadt, Germany). The commercial mycotoxin binder (Toxibond^®^Pro) was purchased from Biomix Co., Ltd. (Sabaneta, Colombia), and yeast extract was purchased from Himedia (Mumbai, India). Malt extract and peptone (Bacto™ Peptone) were purchased from BD Difco™ (Franklin Lakes, NJ, USA). Commercial-grade maltodextrin was purchased from Union Science Co., Ltd. (Chiang Mai, Thailand). Glass beads (3 mm in diameter) were purchased from Paul Marienfeld GmbH & Co., Ltd. KG (Lauda-Königshofen, Germany). All other chemicals used in this study were of analytical grade.

### 2.2. Microorganisms Medium and Culture Conditions

Red yeast (*Sporidiobolus pararoseus* KM281507) was obtained from the culture collection of the Division of Biotechnology, Faculty of Agro-Industry, Chiang Mai University, Chiang Mai, Thailand. Strain KM281507 was maintained in 30% (*v*/*v*) glycerol and stocked at −20 °C. Commercial *S. cerevisiae* (Baker’s yeast, Instant success^®^) was purchased from Lesaffre Group (Lille, France).

The yeast extract–malt extract (YM) medium was used to prepare the seed culture of the red yeast and *S. cerevisiae*. This medium consisted of yeast extract 4 g/L, malt extract 10 g/L and glucose 4 g/L [23]. The initial pH of the YM medium was adjusted to 6.0 by the addition of 1-M H_3_PO_4_. The YM medium was sterilized at 121 °C for 15 min.

Yeast extract–peptone–dextrose (YPD) medium was used to produce the YCs in a 5 L stirred tank bioreactor (MDFT-N-5L, BE Marubishi, Bangkok, Thailand). The YPD medium was composed of yeast extract 10 g/L, peptone 20 g/L and glucose 20 g/L. The initial pH of the YPD medium was adjusted to 6.0 with the addition of 1-M H_3_PO_4_, and the medium was sterilized at 110 °C for 20 min [15].

The basal medium for the production of the RYCs was composed of 1-g/L yeast extract, 5.5-g/L KH_2_PO_4_, 5.3-g/L (NH_4_)_2_SO_4_, 3.7-g/L K_2_HPO_4_, 0.5-g/L MgSO_4_·7H_2_O, 0.2-g/L MnSO_4_·H_2_O, 0.5-g/L NaCl and 50-g/L glucose [22]. To prepare 1 L of basal medium, 1 g of yeast extract, 5.5 g of KH_2_PO_4_, 5.3 g of (NH_4_)_2_SO_4_, 3.7 g of K_2_HPO_4_, 0.5 g of MgSO_4_·7H_2_O, 0.2 g of MnSO_4_·H_2_O and 0.5 g of NaCl were dissolved in 500 mL of distilled water and sterilized at 121 °C for 15 min (solution A). Fifty grams of glucose were separately dissolved in 500 mL of distilled water and sterilized at 121 °C for 15 min (solution B). After cooling, the sterile solutions A and B were aseptically transferred and mixed in the bioreactor. The initial pH was adjusted to 5.63 by the addition of sterile 1-M H_3_PO_4_. This medium was employed to cultivate the RYCs in the 5 L, 30 L and 300 L bioreactors, while the YCs were prepared by transfer of 0.5% (*w*/*v*) Baker’s yeast (*S. cerevisiae*) to 250 mL Erlenmeyer flasks containing 50 mL of YM broth. The flasks were cultivated in an incubator shaker (LSI-3016R, Gyeonggi, Labtech, Korea) at 30 °C, with shaking at 200 rpm until the optical density at 600 nm (OD_600_) reached 1.0. Four flasks of the *S. cerevisiae* culture were aseptically pooled (200 mL) and inoculated into a 5 L stirred tank bioreactor (MDFT-N-5L, BE Marubishi, Bangkok, Thailand) containing 2 L of YPD medium. The aeration rate was maintained at 4 volumes of air (liter) per volume of medium (liter) per minute (vvm), and the agitation rate was 200 rpm. The cultivation was operated at 30 °C for 3 days.

### 2.3. Industrial-Scale Production of Red Yeast Cells

#### 2.3.1. Cultivation of Red Yeast Cells

The production of RYCs was simplified and is shown in Figure 1. To prepare the inoculum, 1 mL of glycerol stock was transferred to a 250 mL Erlenmeyer flask containing 50 mL of YM medium and cultivated in an incubator shaker (LSI-3016R, Labtech, Gyeonggi, Korea) at 24 °C and 200 rpm for 3 days until the OD_600_ reached 1.0. Four flasks of the cultivated red yeast were aseptically pooled (200 mL) and inoculated into a 5 L stirred tank bioreactor (BE Marubishi, Bangkok, Thailand) containing 2 L of basal medium. The aeration rate was maintained at 4 vvm, and the agitation rate was set at 200 rpm. The samples of cultivated red yeast were aseptically sampled from the bioreactor and measured for OD_600_. Cultivation was operated at 24 °C for 3 days or until the OD_600_ reached 1.0, and the cultivated red yeast (2 L) was aseptically transferred into a 30 L stirred tank bioreactor containing 20 L of sterilized basal medium. The 30 L stirred tank bioreactor was equipped with a disc turbine agitator and a baffled cylindrical vessel (BE Marubishi, Bangkok, Thailand). The cultivation condition in the 30 L stirred tank bioreactor was maintained the same as in the 5 L bioreactor for a cultivation time of 24 h. Thereafter, the 24-h red yeast culture (20 L, OD_600_ = 1.0) was aseptically transferred into a 300 L stirred tank bioreactor (BE Marubishi, Bangkok, Thailand) equipped with a disc turbine agitator and a baffled cylindrical vessel containing 200 L of basal medium and cultivated in the same conditions. The effects of pH control regimes, including uncontrolled, semi-controlled and controlled, on the RYC production in the 300 L bioreactor were studied. For the not controlled pH regime, the initial pH was adjusted to 5.63, whereas the controlled pH regime was operated by controlling the pH at 5.63 throughout cultivation period. In the semi-controlled pH regime, the pH was maintained at 5.63 for 48 h and then not controlled until the end of the cultivation period. For the semi-controlled and controlled pH regimes, the pH of the medium was maintained at the set point of 5.63 ± 0.10 by the addition of a sterile KOH solution (1.0 M) using a Multi-channel Bioprocess Control system (MDIAC-S6, BE Marubishi, Bangkok, Thailand).

#### 2.3.2. Downstream Processing of Red Yeast and Yeast Cells

After cultivation in the 300 L stirred tank bioreactor for 3 days using the semi-controlled pH strategy, the cultivated medium containing RYCs was stored at 4 °C for 14 days to allow the autolysis and settling of the RYCs. The supernatant was discarded from the settled RYCs. To prepare the RYCs for glucan content determination, phosphate-buffered saline (pH 7.4) was added, and the RYCs were spray-dried to obtain spray-dried RYCs. For the preliminary in vitro tests of the mycotoxin adsorption capacity, the settled RYCs were mixed to homogeneity with 15% (*w*/*v*) maltodextrin in phosphate-buffered saline (pH 7.4) using an industrial mixer at 70 °C for 15 min. This mixture was spray-dried to obtain encapsulated RYC powder (Petty Patent No.: 18667). The downstream processing of the YCs and the preparation of the spray-dried and encapsulated YCs were prepared under conditions the same as the RYCs.

### 2.4. Analytical Methods

#### 2.4.1. Biomass Measurement

The red yeast biomass was collected from the culture broth and centrifuged at 4430× *g* and 4 °C for 10 min. The cell pellet was washed twice with distilled water, and the dried cell weight (DCW) was determined by drying the cell pellet at 80 °C overnight in a hot air oven and desiccation at 25.0 ± 1.0 °C for 24 h until a constant weight [22].

#### 2.4.2. Extraction and Determination of Carotenoids

The total carotenoids in the RYCs were first extracted by breaking the yeast cells using the modified method of Manowattana et al. [23]. The RYC pellet was harvested from 10 mL of the culture broth by centrifugation at 4430× *g* and 4 °C for 10 min. The RYC pellet was washed twice with *n*-hexane and once with distilled water and extracted in a screw-capped tube containing 10 mL of acetone and 4 g of glass beads in the presence of 100 ppm ascorbic acid by vortexing for 15 min. The broken RYCs were centrifuged at 4430× *g* and 4 °C for 10 min, the clear supernatant collected and flushed with N_2_ gas to ensure complete drying.

Quantitative analysis of the carotenoids was carried out according to Chaiyaso and Manowattana [22]. In brief, the extracted carotenoids were redissolved in 1.0 mL of *n*-hexane, filtered through a 0.2 μm nylon membrane (ALWSCI, Shaoxing, Zhejiang, China) and subjected to high-performance liquid chromatography (HPLC; SCL-10Avp, Shimadzu, Kyoto, Japan) equipped with a C18 column (4.6 × 250 mm; 5 μm, Restek, Bellefonte, PA, USA). The mobile phase consisted of acetonitrile: dichloromethane: methanol (80:10:10, *v*/*v*/*v*), and the flow rate was 1.0 mL/min at 30 °C. The total carotenoid content was detected using a UV–VIS detector (SPD-10Avp, Shimadzu, Kyoto, Japan) operating at 454 nm for 45 min.

#### 2.4.3. Lipid Extraction and Determination

The lipids in the RYCs were extracted using the modified method of Chaiyaso and Manowattana [22]. The RYC pellet was harvested from 20 mL of the culture broth by centrifugation at 4430× *g* and 4 °C for 10 min, washed twice with distilled water and transferred to a screw-capped tube containing 10 mL of a chloroform: methanol mixture (2:1, *v*/*v*) and 4 g of glass beads. The lipid was extracted by vigorously mixing with a vortex mixer for 30 min, sonication at 70 Hz for 30 min and centrifugation at 4430× *g* and 4 °C for 10 min. The clear supernatant was collected, and the organic solvent was removed by evaporation under a pressure at 300 mbar and 25 °C. The lipid production was volumetrically expressed as g/L of culture medium, and the lipid content (%) from the weight of the extracted lipids (g) per dry biomass (g) [22,33].

#### 2.4.4. α- and β-glucan Contents in Yeast Cell Walls

The 1, 3:1, 6-β-glucan of the spray-dried YCs and RYCs (without the addition of maltodextrin) was determined using an enzymatic assay kit (β-glucan assay kit: yeast and mushroom, Megazyme Ltd., Wicklow, Ireland) according to the manufacturer’s instructions (Mushroom and yeast β-glucan assay procedure, K-YBGL 08/18, Megazyme Ltd., Wicklow, Ireland) compared with a standard glucan (control yeast β-glucan preparation, Megazyme Ltd., Wicklow, Ireland). Ninety milligrams of YCs, RYCs and standard glucan were transferred into a 20 × 125-mm tube and solubilization and partial hydrolysis of the total glucan (α-glucan + β-glucan) performed. In brief, 2 mL of ice-cold 12-M H_2_SO_4_ were added to the tubes (with caps) containing spray-dried samples of RYCs or YCs. The tubes were vigorously vortexed and placed in an ice water bath for 2 h with periodic stirring for 10–15 s to complete the dissolution of β-glucan. After the addition of 4 mL of deionized water, the tubes were stirred for 10 s, another 6 mL of deionized water was added and the tubes were stirred for 10 s. The caps were loosened, and the tubes were placed in a boiling water bath. After 5 min, the caps were tightly closed, and the tubes were incubated for another 2 h. After cooling, the caps were carefully loosened. The reaction mixture was transferred to a 100 mL volumetric flask containing 200-mM sodium acetate buffer (pH 5.0), 6 mL of 10-M KOH was added and the reaction volume was adjusted to 100 mL using the 200-mM sodium acetate buffer (pH 5.0). The sample was well-mixed via inversion and transferred to a centrifuge tube. A sample was centrifuged at 1500× *g* for 10 min, and the supernatant was used to determine the total and α-glucan contents.

The total glucan content of 0.1 mL of supernatant was measured by incubation with 0.1 mL of exo-1–3-β-glucanase mixture (20 U/mL) plus β-glucosidase (4 U/mL) in 200-mM sodium acetate buffer (pH 5.0). The reaction mixture was incubated at 40 °C for 60 min, and 3 mL of glucose–oxidase–peroxidase (GOPOD) reagent was added and incubated at 40 °C for 20 min. The absorbance was measured at 510 nm against the reagent blank.

To measure the α-glucan content, 100 mg of the spray-dried YCs, RYCs and standard glucan were stirred with 2 mL of 2-M KOH in an ice bath for 20 min. Eight milliliters of 1.2-M sodium acetate buffer (pH 3.8) and 0.2 mL of amyloglucosidase (1630 U/mL) plus invertase (500 U/mL) were added and incubated in a water bath at 40 °C for 30 min with periodic vortex mixing. The mixture was centrifuged at 1500× *g* for 10 min. Next, 0.1 mL of the supernatant was transferred to a new tube, and 0.1 mL of 200-mM sodium acetate buffer with a pH 5.0 plus 3.0 mL of GOPOD reagent were added, followed by incubation at 40 °C for 20 min. Absorbance was measured at 510 nm against the reagent blank using a spectrophotometer (UV–VIS Genesys 10-s, Thermo Scientific, Loughborough, UK). The β-glucan content was determined by subtracting the α-glucan content from the total glucan content according to the manufacturer’s instructions (mushroom and yeast β-glucan assay procedure, K-YBGL 08/18, Megazyme Ltd., Wicklow, Ireland).

#### 2.4.5. Preparation of Mycotoxin Solutions

Standard mycotoxin solutions were prepared as follows: aflatoxin B1 (AFB1) and T-2 toxin (T-2) were dissolved in acetonitrile (99.9%). Standard ochratoxin A (OTA), zearalenone (ZEA) and deoxynivalenol (DON) were dissolved in methanol (99.9%). Solutions were prepared by dissolving each mycotoxin separately in concentrations in the range of 0.31–2.50 µg/mL.

#### 2.4.6. Preliminary In Vitro Tests for Mycotoxin Adsorption Capacity

The mycotoxin adsorption assay was performed according to the method described by Joannis-Cassan et al. [19] (Figure 2) and the guidelines of the European Food Safety Authority (EFSA). The encapsulated RYCs, YCs and commercial mycotoxin binder (CMB) (50 ± 0.5 mg) were weighted and poured into a 1.5 mL microtube (Eppendorf Safe-lock^®^ tube, Hamburg, Germany) containing 100 µL of mycotoxin solution and 900 µL of phosphate-buffered saline (pH 7.2). The reaction was performed in five replicates. The control was a mycotoxin solution mixed with the same buffer (without adsorbent). All tubes were vortexed and incubated in a thermostatically controlled shaker (AI6R-2 Shel Lab, Sheldon Manufacturing Inc., Cornelius, OR, USA) for 15 min at 37.0 ± 1.0 °C. The reaction tube was centrifuged for 15 min at 9200× *g*, the supernatant filtered through a 0.45 μm nylon membrane and the filtrate was collected for estimation of the mycotoxin concentration using ELISA (RIDASCREEN^®^FAST mycotoxin test kits, R-Biopharm AG, Darmstadt, Germany) according to the manufacturer’s instructions. The limit of detection (LOD) values of the mycotoxin test kit for AFB1, OTA, T-2, ZEA and DON were 1.7, 5, 20, 41 and 200 µg/kg (ppb), respectively. The limit of quantification (LOQ) values of the mycotoxin test kit for AFB1, OTA, T-2, ZEA and DON were 1.7, 5, 50, 50 and 200 µg/kg (ppb), respectively. In brief, 50 µL of filtrate was added to each well. Fifty microliters of conjugate (peroxidase conjugated mycotoxin) and 50 µL of antibody (anti-mycotoxin antibody) were added to each well. The plate was mixed gently and incubated for either 5 min (DON) or 10 min (AFB1, OTA, T-2 and ZEA) at room temperature, and the liquid was discarded from the well. The well was washed twice with 250 µL of phosphate-buffered saline (pH 7.2), and the liquid was removed from the well before the addition of 100 µL of substrate/chromogen. The plate was mixed gently and incubated for either 3 min (DON) or 5 min (AFB1, OTA, T-2 and ZEA) at room temperature in the dark. Finally, 100 µL of stop solution (1-*n* H_2_SO_4_) was added, and the plate was mixed manually. The absorbance of the terminated reaction was measured at 450 nm using a microplate reader (SpectraMax M3; Molecular Devices, San Jose, CA, USA). The absorbance of the zero standard or control (without adsorbents) was used as a reference mycotoxin concentration throughout the experiment [34].

The percentage of adsorbed mycotoxins was calculated using Equation (1):% adsorption = (*C*_ads_/*C*_0_) × 100(1)
where *C*_ads_ is the concentration of adsorbed mycotoxins, and *C*_0_ is the mycotoxin concentration in the supernatant of the control (without adsorbents) [19].

#### 2.4.7. Assessment of Mycotoxin Adsorption Capacity Using In Vitro Gastrointestinal Poultry Model

Broiler chicken feed was formulated and prepared using the recommendations of the National Research Council (NRC) feeding standard. To prepare the mycotoxin-contaminated feed, the feed was sprayed with different mycotoxin concentrations. The mycotoxin concentrations were chosen from the minimum (low) and maximum (high) permitted mycotoxin levels in feed recommended by the European Union (EU). The final mycotoxin concentration was set at two levels: low concentration (AFB1: 0.16, OTA: 0.32, T-2: 0.96, DON: 15 and ZEA: 15 µg/mL) and high concentration (AFB1: 0.32, OTA: 0.64, T-2: 1.92, DON: 30 and ZEA: 30 µg/mL). The sprayed feed was dried at 40 °C overnight in an incubator [35].

The mycotoxin–adsorption capacity of RYCs was tested in a gastrointestinal poultry model compared with CMB. The experiment was carried out in five replicates at 40 °C, which is the core body temperature of poultry [35]. For the first simulation of the gastrointestinal model crop section, three grams of mycotoxin-contaminated feed were mixed with 50 mg of adsorbents (RYCs or CMB) in the 50 mL centrifuge tubes. After that, 10 mL of 0.03-M HCl were added, and the reaction tubes were mixed vigorously. The pH of the reaction mixture was adjusted to 5.2 using 0.2-M NaOH, and the tubes were incubated at 19 rpm in an incubator (AI6R-2 Shel Lab, Sheldon Manufacturing Inc., Cornelius, OR, USA) for 30 min. The proventriculus (enzymatic digestion site) was simulated by the addition of 3000 U/g feed of pepsin (catalog no. 9001-75-6, Sigma-Aldrich, St. Louis, MO, USA) and 2.5 mL of 1.5-M HCl to obtain a pH in the range of 1.4–2.0. The tubes were incubated for 45 min at 19 rpm and 40 °C. Finally, the intestinal section was simulated by the addition of 6.84-mg/g feed 8 × pancreatin (catalog no. 8049-47-6, Sigma-Aldrich, St. Louis, MO, USA). Then, 6.5 mL of 1.0-M sodium bicarbonate (the pH ranged between 6.4 and 6.8) was added, and the tubes were incubated for a further 2 h. The reaction tubes were centrifuged at 2000 rpm for 30 min, and the supernatant was filtered through a 0.45 µm membrane to separate the adsorbents from the aqueous phase, and the filtrate was stored at −20 °C until analysis. The mycotoxin concentration in the filtrate was estimated using ELISA (RIDASCREEN^®^FAST mycotoxin test kit, R-Biopharm AG, Darmstadt, Germany) according to the manufacturer’s instructions, as described in Section 2.4.6. The adsorption percentage was calculated and compared with the value obtained from the control (without adsorbents).

#### 2.4.8. Statistical Data Analysis

The data were analyzed for statistical significance using analysis of variance (ANOVA) followed by Duncan’s multiple range test. The statistical software package SPSS version 20 was used to analyze the experimental data.

## 3. Results

### 3.1. Industrial-Sale Production of Red Yeast Cells

The red yeast *Sporid. pararoseus* KM281507 was cultivated in 5 L, 30 L and 300 L bioreactors. Table 1 shows the production of biomass, lipids and total carotenoids in bioreactors of different scales. The biomass produced in 5 L, 30 L and 300 L bioreactors reached 12.48, 18.02 and 28.70 g/L, respectively. Similarly, the lipids; total carotenoids; the specific growth rate and the biomass, carotenoid and lipid yields increased as the fermentation volume increased. The highest production of all the kinetics parameters from *Sporid.*
*pararoseus* KM281507 was achieved in the 300 L bioreactor. The effects of the pH control strategies, including the uncontrolled, semi-controlled and controlled regimes, on biomass, lipid and total carotenoid production were examined in the 300 L bioreactor (Figure 3). When the pH was not controlled, the yeast strain KM281507 tended to produce more carotenoids than the biomass and lipids. As shown in Figure 3a, the biomass production rate and biomass concentration were the lowest when the pH was not controlled; meanwhile, the biomass concentrations that were produced under the controlled and semi-controlled pH regimes were greater than those from the pH-controlled regime. Likewise, the lipid concentrations from the controlled and semi-controlled pH regimes were almost the same and were significantly higher than those obtained in the uncontrolled pH regime (Figure 3b). On the contrary, the progression of carotenoid production was considerably different. The lowest carotenoid concentration was attained when the pH was controlled, whereas the concentration of the carotenoids produced under the uncontrolled pH regime was the highest (Figure 3c). At the end of the fermentation process, the pH of the medium used in the uncontrolled and semi-controlled pH regimes was 2.74 ± 0.23 and 2.89 ± 0.24, respectively. In contrast, when the pH was controlled (pH 5.63), biomass and lipid production were enhanced, whereas carotenoid production was relatively low. The biomass obtained via pH control was approximately three times higher than that in the system with an uncontrolled pH. Interestingly, the biomass obtained from the system with a semi-controlled pH was slightly lower than that in the controlled system, whereas the lipid production was similar. Although the carotenoid production rate of the semi-controlled pH regime was slightly lower than that of the uncontrolled pH, the carotenoid yield obtained from the semi-controlled pH at the end of the cultivation period was similar (Figure 3).

RYC production was performed via the cultivation of strain KM281507 in the 300 L bioreactor under a semi-controlled pH regime. The spray-dried RYCs (without the addition of maltodextrin) were analyzed to determine the glucan content compared to the standard glucan and the spray-dried YCs. Overall, the β-glucan content was substantially higher than the α-glucan content (Table 2). Based on the data from Table 2, the total glucan and β-glucan contents in the YCs were higher than those obtained from the RYCs.

### 3.2. Mycotoxin Adsorption Capacity

#### 3.2.1. Preliminary In Vitro Tests for Mycotoxin Adsorption Capacity

The mycotoxin adsorption capacity of the encapsulated RYCs and YCs was investigated using different mycotoxin concentrations of 0.31, 0.62, 1.25 and 2.50 µg/mL. The capacity was compared to that of the YCs and CMB. As depicted in Table 3, the CMB had the highest capacity (> 80%) to bind AFB1, whereas the capacities of the RYCs and YCs to bind 2.5-µg/mL AFB1 were comparable. When the AFB1 concentration was 0.31–1.25 µg/mL, the RYCs were able to bind AFB1 better than the YCs could. For the ZEA experiments, the potential of the three mycotoxin adsorbents was similar when using 0.31 µg/mL of ZEA. However, at higher ZEA concentrations (0.62–2.50 µg/mL), the mycotoxin adsorption capacity of the CMB and the YCs were the highest. The capacity of the YCs and CMB to adsorb DON were almost the same compared to those of ZEA. In addition, the RYCs had the highest potential to adsorb OTA and T-2 at all of the concentrations tested. However, according to the recommendation of the EFSA, the adsorption level of T-2 and DON were not high enough (at least 20%). Therefore, the mycotoxin adsorption capacity was further tested in the following section.

#### 3.2.2. Assessment of Mycotoxin Adsorption Capacity Using In Vitro Gastrointestinal Poultry Model

The capacity of RYCs to adsorb mycotoxins was examined by simulating a gastrointestinal model of poultry, and the results are presented in Figure 4. At low mycotoxin concentrations, the capacities of the RYCs to adsorb ZEA, AFB1, OTA, DON and T-2 were 99.00, 93.00, 79.10, 72.87 and 59.10%, respectively (Figure 4a). Similarly, the adsorption percentages of CMB were 99.30, 91.00, 75.30, 74.07 and 58.60% when ZEA, AFB1, OTA, DON and T-2 were tested. Compared to CMB, the adsorption capacities of the RYCs were not significantly different, and both adsorbents effectively sequestered mycotoxins. Regarding the high mycotoxin concentrations, RYCs exhibited a higher capacity to adsorb AFB, OTA and DON than CMB did (Figure 4b).

## 4. Discussion

Regarding economic and industrial feasibility, scaling up RYC production is a necessity. In the present study, the biomass, lipid and carotenoid levels obtained from the 30 L and 300 L bioreactors were higher than those obtained from the 5 L bioreactor (Table 1). This can be explained by the greater amount of oxygen transfer, resulting from the disc turbine agitator and baffled cylindrical vessel in the 30 L and 300 L bioreactors. The turbulence and gas flow rates are critical factors for heat and oxygen transfer, impacting aerobic processes such as yeast cell growth [36]. Moreover, the aeration rate is a crucial factor that affects biomass and lipid production, as well as carotenogenesis [37]. In a previous study, *Rhodosporidium toruloides* RP15 was cultured in a stirred tank bioreactor. Biomass, lipid and carotenoid concentrations of 37.5, 22.4 and 10.7 g/L were obtained [38]. *Rhodotorula taiwanensis* AM2352 could convert corncob hydrolysate into a 18.7-g/L biomass with 60.3% (*w*/*w*) of lipids under batch cultivation in a 5 L bioreactor [39]. Similarly, the lipid production by *Rhodotorula mucilaginosa* IIPL32 in a 70 L pilot-scale bioreactor was enhanced to 1.83 g/L [36].

The effect of a controlled pH regime was examined in the 300 L bioreactor, and it revealed that the pH value has a major influence on cell growth and the lipid and carotenoid yields. During the cultivation, the biomass increased, and the lipids and carotenoids were synthesized and accumulated in the yeast cells [36]. The carotenoids that were produced by oleaginous red yeast were stored and protected in the lipid droplets [40]. When the pH was not controlled, carotenoid production was promoted, whereas biomass and lipid production was impeded. Lowering the pH can cause oxidative stress, which enhances carotenoid biosynthesis [41]. Our results showed that the biomass obtained via the controlled pH regime was approximately three times higher than that in the uncontrolled pH regime (Figure 3). However, regulating the pH value throughout the fermentation period may not be desirable for carotenoid production [41], because fatty acid and lipids are growth-associated compounds, whereas carotenoids are synthesized during the exponential and stationary phases [42]. Under the semi-controlled pH condition, the pH was maintained at 5.63 for 48 h and was then uncontrolled until the end of the cultivation period. After 48 h, the pH measured from this condition sharply dropped from 5.63 to 2.89. Consequently, the carotenoid production also increased as the pH of the cultivation medium decreased. According to previous reports, the decrease in the pH (lower than 4.32) is one of the main reasons for the enhanced β-carotene, total carotenoids and astaxanthin production in red yeasts *Sporid. pararoseus* KM281507, *Xanthophyllomyces dendrorhous* and *Sporid. salmonicolor* CBS 2636 [22,43,44]. Acidic conditions could induce *Rhodosporidium paludigenum* KM281510 carotenoid accumulation, while neutral and basic conditions could promote biomass and lipid production [33]. Hence, the semi-controlled pH regime was developed to simultaneously obtain high yields of biomass, lipids and carotenoids from red yeast. As shown in Figure 3 and Table 1, the biomass, lipid and total carotenoid concentrations obtained from the semi-controlled system were relatively high.

In general, the cells walls of yeast contain 50–60% β-glucan and 40% mannoproteins [45]. β-glucan is a polysaccharide that consists of a β-D-glucose unit and is found in yeasts and mushrooms linked with β-1, 3 and β-1, 6 glycosidic bonds. The biosynthesized β-glucan can increase the cell wall thickness, resulting in cell wall protection. The conventional yeast *S. cerevisiae* is an important source of β-glucan [46,47,48]. Yeast β-glucans are generally recognized as safe (GRAS) and are widely used as food ingredients [49]. The β-glucan content of the YCs that were analyzed in this study was approximately 9% (*w*/*w*). These results are in line with previous studies in which the β-glucan content of the *S. cerevisiae* cell wall was in the range of 7.7–18% [45,49]. The total and β-glucan contents of the RYCs were lower than those of the YCs (Table 2). Glucan, lipid and carotenoid production depend on multiple factors, including yeast strains, C/N ratios and cultivation conditions [50,51]. The coproduction of β-glucan and lipids has been investigated by employing carotenogenic basidiomycetes [50]. The β-glucan contents of the genera *Rhodotorula*, *Cystophilobasidium*, *Sporobolomyces* and *Phaffia* were analyzed compared to those of *S. cerevisiae*, the ascomycetes yeast. The results showed that the β-glucan contents of *S. cerevisiae* were higher than those of carotenogenic basidiomycetes. Another study noted that polysaccharide synthesis could suppress lipid and carotenoid synthesis in the *Rhodotorula* sp. [52]. However, the relationship between carotenoid, lipid and β-glucan synthesis has not been clearly elucidated.

Mycotoxins are fungi-produced secondary metabolites that have toxic effects on humans and animals [35]. Since different fungal species can grow and synthesize various mycotoxins under comparable environmental conditions, it is difficult to find cereal grains that have been contaminated with a single mycotoxin [34,35]. Furthermore, animal diets generally contain numerous ingredients, each with a distinct mycotoxin or multiple mycotoxins. Therefore, organic adsorbents are commonly used to prevent mycotoxicosis [34]. However, their abilities are relatively specific depending on the type of mycotoxin. For example, adsorbents with a high capacity to bind a particular mycotoxin might not bind to other mycotoxins [53]. During the last decade, many studies have focused on the potential of the yeast cell wall to adsorb mycotoxins and form an adsorbent–toxin complex [20]. Desirable functions include the potential to bind, inactivate and transport the mycotoxin throughout the gastrointestinal tract without harmful effects [54]. In the present study, RYCs, an emerging organic adsorbent, exhibited the in vitro capability to sequester various mycotoxins. However, the in vitro experimental conditions alone cannot sufficiently confirm the capability of these cells, making it difficult to compare our studies to other reports due to the differences in the pH and reaction time. The standard protocol described in the present study is based on the recommendations provided for adsorption studies on YCs. YCs are widely used as adsorbents for mycotoxin removal [19]. As far as we know, this is the first study in which RYCs were tested to determine their binding capability for use as an adsorbent against mycotoxins. The capacity of the RYCs to adsorb OTA and T-2 was higher than that of the CMB and YCs (Table 3). Gallo and Masoero [55] reported that the adsorption percentages of AFB1 by YCs were in the range of 32–54%, whereas the adsorption percentages of ZEA ranged from 22 to 62% when the same mycotoxin concentrations were investigated. In a previous study, it was easier for OTA to be adsorbed than it was for T-2 and DON to be adsorbed [19].

The chemical interactions between binders and toxins have not been clearly described. The adhesion or sequestration between YCs and mycotoxins is due to cell wall components such as glucomannans, mannans and β-glucans [32,56,57]. The possible explanation for this may be the presence of β-glucans in YCs and RYCs. The main component of the cell wall of red yeast is β-D-glucan, which accounts for 5.7% of the dry weight of red yeast (Table 2). The helix of β-(1,6)-D-glucan from the YCs is the key factor for mycotoxin adsorption [56]. When their helical conformation relaxes, the distance between the β-D-glucan molecules is also increased [15,56,58]. Yiannikouris et al. [16] reported that the laminarin molecule, a pure β-(1,3)-D-glucan, possesses β-D-glucopyranose residue single-helix conformation [16]. This conformation is adequate for interactions between β-glucans and ZEA. Moreover, these interactions are consistent due to the stabilization of β-(1,6)-D-glucan side chains, most likely because more than one mechanism is involved in the interactions between mycotoxins and β-glucans. These interactions might be due to hydrogen bonds and Van der Waals forces [59]. The factors affecting the binding capacity vary because of the intrinsic characteristics of both mycotoxins and binders, including the types, chemical structure and polarity of mycotoxins, as well as the cell wall components and size of the yeasts [56,57]. Although, RYCs have a higher adsorption capacity than that of YCs and CMB, the absorption percentages of T-2 and DON were in the lower range, because T-2 and DON are very difficult to adsorb [17,60]. The structure of T-2 is similar to that of DON, but its toxicity is higher than DON [61]. The cell walls of yeast are mainly composed of organic substances posing as various functional groups and hydrophobic adsorption sites [62]. The adsorption of T-2, a nonpolar mycotoxin, might be due to the interaction with hydrophobic compounds [63]. Meanwhile, the polysaccharide components of the YCs are responsible for OTA adsorption [27]. Generally, the adsorption capacity in the in vitro gastrointestinal model was higher than that obtained from preliminary tests (without pH control). When the pH was lower than 5.0, the conformation of the mycotoxins and the interactions between RYCs and mycotoxin changed due to an increase in van der Waals forces and hydrogen bonding [16,54]. Another possible reason is the change in the pH to 6.5, which can increase mycotoxin detachment [61]. Acidic pHs of 2.5 and 6.5 could preserve the helical shape of β-glucan and maintain the docking site of protonated OTA [54]. Before validation using an in vivo model, it is essential to simulate an in vitro gastrointestinal model to understand the factors affecting the adsorption capacity of different adsorbents.

## 5. Conclusions

The industrial-scale production of RYCs in a 300 L bioreactor under a semi-controlled pH regime resulted in 28.70 ± 2.34-g/L biomass, 8.67 ± 0.64-g/L lipids and 96.10 ± 3.67-mg/L total carotenoids. The produced RYCs contained 5.73 ± 0.12% (*w*/*w*) β-glucan. They had the highest capacity to bind with OTA and T-2 at concentrations of 0.31–1.25 and 0.31–2.5 µg/mL, respectively. Furthermore, the mycotoxin adsorption capacity, which was examined through an in vitro gastrointestinal poultry model, revealed that the adsorption capacities of the RYCs and CMB were comparable. Apart from the mycotoxin binding capacity, RYCs also contain carotenoids and lipids, making them suitable for use as an industrially feasible feed additive.

## 6. Patents

International Patent Classification: C12N 1/04; Petty Patent No.: 18667.

## Figures and Tables

**Figure 1 jof-08-00353-f001:**
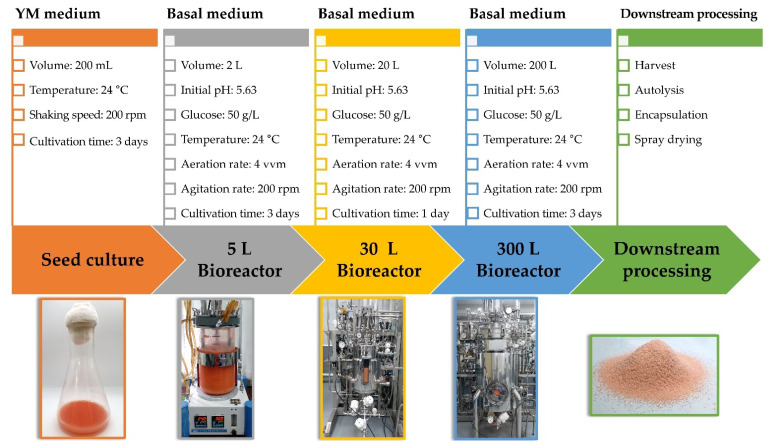
Flow diagram showing the industrial-scale production of red yeast cells.

**Figure 2 jof-08-00353-f002:**
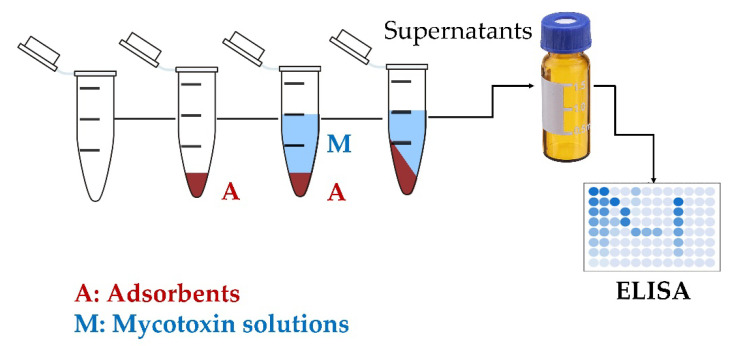
Schematic diagram of the preliminary in vitro tests: A = adsorbents; M = mycotoxin solutions.

**Figure 3 jof-08-00353-f003:**
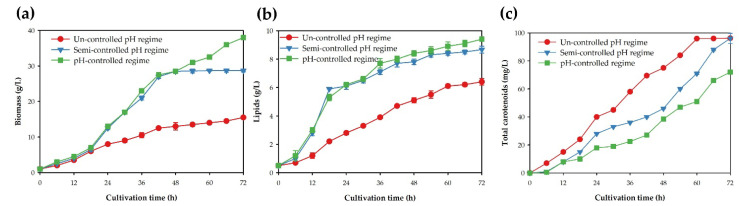
Effects of different pH control regimes on the biomass (**a**), lipid (**b**) and total carotenoid production (**c**) of *Sporid. pararoseus* KM281507 in the 300 L bioreactor.

**Figure 4 jof-08-00353-f004:**
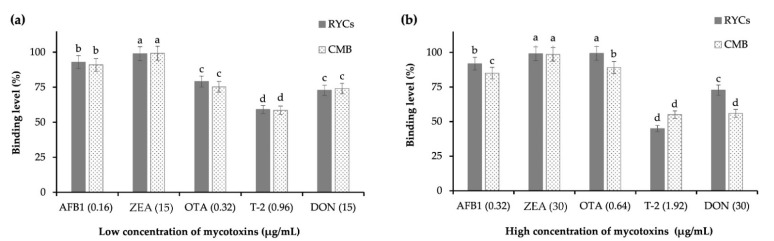
Comparison of the adsorption percentages of five mycotoxins at low (**a**) and high (**b**) concentrations (µg/mL). Bars show the mean of five replicates (*n* = 5), and the error bars are the standard error of the mean. Each value was compared between mycotoxin types. Bars with same letters (a–d) are not significantly different (*p* > 0.05). RYCs: red yeast cells; CMB: commercial mycotoxin binder.

**Table 1 jof-08-00353-t001:** Production of biomass, lipids and total carotenoids from the red yeast *Sporid. pararoseus* KM281507 under cultivation in 5 L, 30 L and 300 L stirred tank bioreactors.

Kinetic Parameters	5 L Bioreactor (120 h)	30 L Bioreactor (72 h)	300 L Bioreactor (72 h) *
Biomass (g/L)	12.48 ± 0.93 ^c^	18.02 ± 1.23 ^b^	28.70 ± 2.34 ^a^
Lipids (g/L)	3.78 ± 0.12 ^c^	4.92 ± 0.28 ^b^	8.67 ± 0.64 ^a^
Total carotenoids (mg/L)	60.26 ± 1.34 ^c^	68.21 ± 2.11 ^b^	96.10 ± 3.67 ^a^
Specific growth rate (µ, h^−1^)	0.10 ± 0.01 ^c^	0.25 ± 0.06 ^b^	0.39 ± 0.09 ^a^
Yield of biomass (Y_X/S_, g/g)	0.25 ± 0.02 ^c^	0.36 ± 0.02 ^b^	0.57 ± 0.05 ^a^
Yield of carotenoids (Y_C/X_, mg/g)	4.83 ± 0.25 ^a^	3.78 ± 0.14 ^b^	3.35 ± 0.15 ^c^
Yield of lipids (Y_L/X_, g/g)	0.30 ± 0.01 ^a^	0.27 ± 0.00 ^b^	0.30 ± 0.00 ^a^

Note: Values are presented as the mean ± standard deviation. Data with the same superscripts in the same row are not significantly different at *p* ≤ 0.05 (kinetic parameters were compared among the different scales). The level of significance was tested by Duncan’s multiple range test at *p* ≤ 0.05. * The 300-L bioreactor was subjected to the semi-controlled pH regime.

**Table 2 jof-08-00353-t002:** Glucan contents of standard glucan, yeast (*S. cerevisiae*) and red yeast (*Sporid. pararoseus*) cells.

Samples	Total Glucan (%, *w*/*w*)	α-glucan (%, *w*/*w*)	β-glucan (%, *w*/*w*)
Standard glucan *	40.53 ± 0.42 ^a^	0.62 ± 0.00 ^c^	39.91 ± 0.31 ^a^
Yeast cells (YCs) **	11.33 ± 0.11 ^b^	2.16 ± 0.21 ^a^	9.18 ± 0.18 ^b^
Red yeast cells (RYCs) **	7.40 ± 0.13 ^c^	1.63 ± 0.10 ^b^	5.73 ± 0.12 ^c^

Note: Values are presented as the mean ± standard deviation. Data with the same superscripts (a–c) in the same column are not significantly different (different glucan contents were compared between samples). The level of significance was tested by Duncan’s multiple range test at *p* ≤ 0.05. * Standard glucan refers to the control yeast β-glucan preparation from Megazyme Ltd. (Ireland). ** Yeast and red yeast cells were obtained from the spray-drying process without the addition of maltodextrin, as described in Section 2.3.2.

**Table 3 jof-08-00353-t003:** Adsorption levels of the mycotoxins adsorbed in vitro by different adsorbents.

Adsorbents	Mycotoxin Concentration (µg/mL) and Adsorption Levels (%)			
	2.50	1.25	0.62	0.31
Aflatoxin B1 (AFB1)				
Yeast cells (YCs)	12.27 ^b^	11.90 ^c^	25.96 ^c^	51.33 ^c^
Red yeast cells (RYCs)	11.30 ^b^	24.67 ^b^	50.00 ^b^	76.30 ^b^
Commercial mycotoxin binder (CMB)	85.13 ^a^	87.00 ^a^	87.33 ^a^	94.53 ^a^
SEM	12.227	11.602	8.935	6.268
*p*-value	<0.001	<0.001	<0.001	<0.001
Zearalenone (ZEA)				
Yeast cells (YCs)	98.70 ^a^	98.46 ^a^	97.93 ^a^	95.27 ^a^
Red yeast cells (RYCs)	86.67 ^b^	87.80 ^b^	82.93 ^b^	99.50 ^a^
Commercial mycotoxin binder (CMB)	100.00 ^a^	99.43 ^a^	99.53 ^a^	99.27 ^a^
SEM *	2.381	1.923	2.674	1.520
*p*-value	0.005	<0.001	<0.001	0.504
**Ochratoxin A** (**OTA**)				
Yeast cells (YCs)	24.93 ^a^	35.36 ^c^	64.13 ^b^	29.00 ^c^
Red yeast cells (RYCs)	22.66 ^a^	49.86 ^a^	65.53 ^a^	61.03 ^a^
Commercial mycotoxin binder (CMB)	24.56 ^a^	40.53 ^b^	59.70 ^c^	54.80 ^b^
SEM *	0.446	2.130	0.899	4.946
*p*-value	0.054	<0.001	<0.001	<0.001
T-2 toxin (T-2)				
Yeast cells (YCs)	8.40 ^a^	12.87 ^b^	16.00 ^b^	12.86 ^b^
Red yeast cells (RYCs)	8.33 ^a^	19.23 ^a^	32.93 ^a^	33.03 ^a^
Commercial mycotoxin binder (CMB)	4.43 ^b^	8.03 ^c^	13.20 ^c^	10.50 ^b^
SEM *	0.714	1.639	3.093	3.602
*p*-value	0.004	<0.001	<0.001	<0.001
Deoxynivalenol toxin (DON)				
Yeast cells (YCs)	9.70 ^a^	12.50 ^a^	10.43 ^a^	18.20 ^a^
Red yeast cells (RYCs)	6.73 ^b^	4.80 ^c^	5.33 ^b^	17.73 ^a^
Commercial mycotoxin binder (CMB)	10.10 ^a^	10.56 ^b^	10.80 ^a^	19.23 ^a^
SEM *	0.557	1.163	0.900	0.364
*p*-value	0.001	<0.001	<0.001	0.249

Note: The values are given as the percentage of mycotoxins adsorbed into each adsorbent. Values are means of five replicates (*n* = 5). The level of significance was tested by Duncan’s multiple range test at *p* ≤ 0.05. Data with the same superscripts (a–c) are not significantly different (different adsorption levels were compared between adsorbents). * SEM refers to standard error of the mean.

## Data Availability

The petty patent No.: 18667 has been submitted to Department of Intellectual Property (DIP), Ministry of Commerce, Nonthaburi, Thailand, and is available on request from the corresponding author (thanongsak.c@cmu.ac.th).

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
