# Peer review of "Industrial-Scale Production of Mycotoxin Binder from the Red Yeast Sporidiobolus pararoseus KM281507"

_jof, 2022, doi:10.3390/jof8040353_

Round 1

Reviewer 1 Report

This study investigated the industrial feasibility of the large-scale production of RYCs along with its ability as a mycotoxin binder. Under semi-controlled pH condition in the 300-L bioreactor, 28.70 g/L biomass, 8.67 g/L lipids, and 96.10 mg/L total carotenoids were obtained, and the encapsulated RYCs contained 5.73% (w/w) β-glucan. This encapsulated RYC was in vitro preliminary tested for its mycotoxin adsorption capacity, including aflatoxin B1 (AFB1), zearalenone (ZEA), ochratoxin A (OTA), T-2 toxin (T-2), and deoxynivalenol (DON).

I would recommend acceptance after minor revisions .

1. Part 4 should be concise and comprehensive. Please rephrase.

2. Keywords : Five words are the best.

3.  Line 124-130:  Why this paragraph use  italic format?

4. There is a nalnk space between the number and the unit, please check the whole text. (e.g. L 313)

5. The experimental mechanism is not clearly articulated.

Author Response

Responses to the comments of the Reviewer 1

We would like to thank the Reviewer 1 for valuable comments and suggestions of the manuscript. We have read your comments carefully and made the correction. We hope that this revised version is suitable for publication. We have responded to your comments using the “Track Changes” function and outlined each change made (point by point) as the attachment.

Reviewer 2 Report

This paper presents the industrial scale production of a red yeast as well as the binder ability of this microorganism on different mycotoxins common in. poultry feed.

Due to the economical and sanitary importance of mycotoxins as feed contaminants, this subject is of interest.

The paper seems well and clearly written.

I nevertheless have some few questions to be answered before publication.

  • Lines 107 and 112: why sterilization procedures (temperature) were different?
  • Lines 113-117: was the resulting medium sterilized?
  • Lines 1122-130: why is this paragraph in italic? It does not seem well placed in the manuscript and redundant with the following. Since culture conditions for YC and RCY were very closed, probably they can be grouped.
  • Lines 150 and later in the text: change “un-control, semi-control and control” with “uncontrolled, semi-controlled and controlled conditions”
  • Paragraph 2.4.4: why having measured beta and alpha-glucan on encapsulated yeasts cells? What control was used (capsules with no YC)?
  • Since encapsulation was done and since it appears that RYC could be sued in this form, were they used to test binding efficiency or were the test done directly using unencapsulated YC? In this later case, what is the interest of encapsulation?
  • Preliminary binding tests, line 266: why working with a so small weight of binder (50 mg), which can be difficult to measure and manipulate?
  • Line 274-275: give analytical performances of kits used (LOQ, LOD, range of detection: did you need any dilution before measurement?).
  • Lines 293 -294: what are the corresponding toxins for each given low and high concentrations? How were these values chosen?
  • Paragraph 2.4.7: on what part (supernatant-binders) were done toxine measurement. Add the method used for toxin measurement (even if the same than previously, then say it).
  • Line 383: already said line 378 (better binding of AFB1)
  • Table 3: you could put in bold the best results obtained with RYC for OTA and T-2 in order to highlight them?

Author Response

Responses to the comments of the Reviewer 2

We would like to thank the Reviewer 2 for valuable comments and suggestions of the manuscript. We have read your comments carefully and made the correction. We hope that this revised version is suitable for publication. We have responded to your comments using the “Track Changes” function and outlined each change made (point by point) as the attachment.

Reviewer 3 Report

Dear Author,

Please find the comments in attached file. 

Delete the make of common instruments through out the manuscript. 

Reduce the keywords no.

There were one paragraph in italic format. Change it to normal 

Author Response

Responses to the comments of the Reviewer 3

We would like to thank the Reviewer 3 for valuable comments and suggestions of the manuscript. We have read your comments carefully and made the correction. We hope that this revised version is suitable for publication. We have responded to your comments using the “Track Changes” function and outlined each change made (point by point) as the attachment.

This manuscript is a resubmission of an earlier submission. The following is a list of the peer review reports and author responses from that submission.

Round 1

Reviewer 1 Report

This article describes the industrial-scale production of a Yeast (Sproridiobolus pararoseus KM281507) and its interest due to mycotoxin binding ability of yeast cells walls.

Due to the sanitary importance of mycotoxin contamination of foods and feeds and the need to find strategies to limit their toxic effects, especially in animals, this paper is of interest. It is also interesting to see that this product is quite efficient against T2 toxin and OTA, for which there is at the moment only few identified binders.

If the production of the yeast is quite well described, I have some remarks on the methodology used to test binding ability. Some clarification shall be brought before the publication of this work.

Here are these remarks according to their order in the text

Introduction: what is the target species for this feed additive? Indeed, the fact that the yeast produce carotenoids may lead to color change of the lipids of fed animals which could be a problem. It is also important to clarify that for the binding testing methodology: monogastric vs ruminants and corresponding intestinal pH.

Material and methods

Lines 82-83: should be placed in the following paragraph (Microorganisms…)

Line98: how pH was adjusted?

Line 104: glucose was separately dissolved: what concentration?

Line 107: “commercial production” or “industrial production”?

Lines 110-113: first step is done with 50 mL then 200 mL are inoculated… so does it means that 4 flasks are pooled together?

Lines 117-121: was the OD monitored as done at the beginning to control yeast development at each step?

Lines 127-128: how pH was adjusted?

Line 132: why 3 days of incubation? Was the OD measured? Were the yeast counted to identify this time as relevant.

Line 148: time and temperature used for desiccation?

Paragraph 2.4.3: if the extraction of lipids is described, it seems that the method for determination is missing.

Paragraph 2.4.6: this is the part that needs some precision. Indeed, when testing binders, it is necessary to be as close as possible to the pH of the intestine of target animals and its modification during digestion. Binding of mycotoxins can strongly be affected by pH and its consequence of the ionization of the molecule. So, a toxin bound at acidic pH could be released later in the digestive tract when pH turns to neutral or the contrary. That is why different pH shall be tested.

There are also some clarifications about the control and the calculation of binding ability. Binding ability shall be expressed compared to the measurement of the toxins incubated with no binding agent (and not compared to the theorical concentration due to the loss of some molecules during analytical procedure that is not linked to the binding process).

Some recommendations were published by EFSA about the best practices to be used in the in vitro testing of binders. Authors should read that and explain and discuss their experimental choices (See EFSA : review of mycotoxin detoxifying agents used as feed additives: mode of action, efficacy and feed/food safety, p131-132). The most important points are: simulation of the gastrointestinal conditions, concentration of the binders vs the toxins, calculation of binding ability, analytical performances of the method used for quantification of mycotoxins.

Figure 3: since pH control, at least partial one is of great influence on the results, the pH in uncontrolled conditions shall be given. On figure 3C, how do the authors explain that in semi-controlled pH conditions, the different is greater with controlled pH at the beginning rather than at the end? This is different from the two other parameters shown here (figure 3a and b).

Paragraph 3.2: authors shall precise what are the corresponding concentrations of the tested toxins in feed (theorical calculation). Does the concentration tested here correspond to highly contaminated feeds or, contrary to feeds below EU regulation regarding tested compounds?

For binding assays, were the different binders used as the same concentration? How was it chosen?

Discussion: since the main reason of developing the industrial-scale production is the mycotoxin-binding ability of the yeast walls, shouldn’t it be more logical to first test this binding ability then develop industrial scale and finally verify that the product still works?

Lines 359-374: mostly generality. Authors shall probably focus their discussion on the possible mechanism of action, especially since their binder can blocks mycotoxins that are poorly bound by others (T-2 and OTA).

Line 367: “recently” is probably not adapted since binders (including Yeast walls) are searched for decades (see ref 15 and 16).

Author Response

We would like to thank the Reviewers for your valuable comments and suggestions of the manuscript. We have read your comments carefully and made the correction. We have responded to your comments using the “Track Changes” function and outlined each change made (point by point) as attached files.

Reviewer 2 Report

This research showed that the industrial-scale production of RYC in the 300-L bioreactor under semi-controlled pH values was 28.70 ± 2.34 g/L biomass, 8.67 ± 0.64 g/L lipids, and 96.10 ± 3.67 mg/L total carotenoids. The produced RYC contained 5.73 ± 0.12% (w/w) β-glucan. It had the highest  capacity to bind with OTA and T-2 at concentrations of 0.31–1.25 and 0.31–2.5 µg/mL,  respectively. Apart from the mycotoxin binding ability, RYC also contains carotenoids and lipids, making it as industrially feasible feed additive. Researchers in the field of mycotoxins and industrial production may be interested in the study. I would recommend minor revisions before acceptance. 

Author Response

(The authors gave the same response as above.)

Round 2

Reviewer 1 Report

The authors strongly improved the manuscript and brought relevant answers to all remarks done previously. So, to my point f view, the paper is now acceptable for publication in Journal of Fungi.